# Selection of Organic Fluid Based on Exergetic Performance of Subcritical Organic Rankine Cycle (ORC) for Warm Regions

Muhammad Ehtisham Siddiqui [1,*], Eydhah Almatrafi [2], Usman Saeed [3] and Aqeel Ahmad Taimoor [3]

1 Mechanical Engineering Department, King Abdulaziz University, Jeddah 21589, Saudi Arabia
2 K. A. CARE Energy Research and Innovation Center, Centre of Excellence in Desalination Technology, Department of Mechanical Engineering, Faculty of Engineering-Rabigh, King Abdulaziz University, Jeddah 21589, Saudi Arabia
3 Department of Chemical and Materials Engineering, Faculty of Engineering, King Abdulaziz University, Jeddah 21589, Saudi Arabia
* Correspondence: mesiddiqui@kau.edu.sa or ehtisham.siddiqui@gmail.com; Tel.: +966-55-218-4681

**Abstract:** The organic Rankine cycle (ORC) exhibits considerable promise in efficiently utilizing low-to-medium-grade heat. Currently, there is a range of organic fluids available in the market, and selecting the appropriate one for a specific application involves considering factors such as the cycle's thermodynamic performance, plant size, and compatibility with turbomachinery. The objective of our study is to examine the exergetic performance of the ORC with internal heat regeneration. We analyze 12 different organic fluids to evaluate their suitability based on parameters like exergy efficiency and heat exchange area requirements. Additionally, we investigate the need for internal heat regeneration by comparing the overall exergy performance with a simpler ORC configuration. To ensure broad applicability, we consider source temperatures ranging from 150 to 300 °C, which are relevant to industrial waste heat, geothermal sources, and solar energy. For each case, we calculate specific net power output and the UA value (heat exchanger conductance) to gain insights into selecting the appropriate organic fluid for specific source temperatures. Cyclohexane, benzene, isopropyl alcohol, and hexafluorobenzene show poor exergy efficiency due to their high boiling points. Pentane and cyclopentane provides the highest exergy efficiency of 62.2% at source temperature of 300 °C, whereas pentane is found to be the most suitable at source temperatures of 200 and 150 °C with exergy efficiency of 67.7% and 61.7%, respectively. At 200 °C source temperature, RE347mcc achieves 65.9% exergy efficiency. The choice of organic fluid for a given heat source is highly influenced by its critical properties. Moreover, the normal boiling temperature of the organic fluid significantly impacts exergy destruction during the condensation process within the cycle.

**Keywords:** exergy analysis; internal heat recovery; organic fluids; organic Rankine cycle; thermodynamic assessment; warm region

## 1. Introduction

The reality of climate change posing a threat to our planet is undeniable [1]. Earthquakes of alarming magnitudes are occurring more frequently than ever before. Areas that were once cold are now experiencing warm or hot climates. Cities are grappling with heavy rainfall, intense storms, and floods that were previously unheard of. One of the main causes behind this concerning climate change is the increased reliance on fossil fuels for power generation. The rapid growth of industrialization necessitates a shift in our approach to generating electricity. Using the conventional steam Rankine cycle for power generation becomes impractical when utilizing low-grade renewable thermal resources such as geothermal and solar energy, or waste heat. This is because steam Rankine cycles are inefficient for low-temperature sources [2,3]. To address this challenge, an alternative option emerged: replacing water with a suitable working fluid for low-temperature sources. This led to the development of the organic Rankine cycle (ORC). Operating on the same

principle as the steam Rankine cycle, the ORC employs organic fluids with significantly lower boiling points than water. The ORC technology shows promise for power generation from low-temperature sources due to its simplicity, cost-effectiveness, and low maintenance requirements [4,5]. Consequently, the ORC has garnered significant attention from relevant authorities in recent years. The global ORC market has witnessed substantial growth, with a cumulative installed capacity of 4.5 GW as of 2020, representing a nearly 40% increase since 2016 in terms of capacity, and a 47% increase in the number of installed plants worldwide [6].

Because the ORC is proposed for low-temperature applications, its thermal efficiency is naturally low [7,8]. Moreover, the selection of organic fluid for the cycle also has a significant impact on the overall performance of the cycle; this is due to the thermophysical properties of the organic fluid, for example, normal boiling temperature, critical pressure and temperature, molar mass, latent heat, saturation properties [9]. Appropriate selection of the organic fluid for a given source temperature is crucial to efficiently operate the cycle; therefore, most published research on ORC in the past decade is focused on the selection of organic fluid [9,10]. Some of them offer guidelines on the selection criteria of a proper organic fluid for a range of source temperatures and source types, for example, low-grade waste heat [11,12], geothermal [13,14], and solar [15,16].

Recently, binary mixtures of organic fluids have also caught a lot of attention from the scientific community and roved to be a potential working fluid in ORC instead of pure organic fluid [17–21]. Pure organic fluids incur substantially large exergy losses in the evaporator and condenser, which is mainly due to the isothermal phase change process. Binary mixtures aim to reduce these irreversibilities by matching the temperature profiles of the heat source in the evaporator and heat sink in the condenser [18,19]. This improves the overall thermodynamic performance of the cycle. However, one of the main shortcomings of binary mixtures is their degraded heat transfer characteristics compared to their pure constituents. This subsequently demands larger heat exchangers, which ultimately increase the cost and size of the plant [19]. Despite the improved overall thermodynamic performance of ORC using binary mixtures, Wu et al. found smaller net power output per unit UA (heat exchanger conductance) with binary mixture compared to cases studied with pure fluids [22]. A thermo-economic study by Oyewunmi and Markides also suggests an increase in plant cost by 14% if operating with binary mixtures, which are due to the requirement of larger condensers, evaporators, and expanders when compared to pure organic fluid [23].

Considering the environment, researchers have studied the various low global warming potential (GWP) organic fluids for ORC [24–26]. Recently, Bahrami et al. reviewed various ecologically friendly organic fluids for ORC applications [27]. Hydrocarbons (HCs) and hydrofluorocarbons (HFCs) are good candidates for ORC due to their good thermophysical properties. HCs despite being highly flammable are appealing due to zero ozone depletion potential (ODP) and low GWP. HFCs are commonly less dense than HCs, which results in low specific net power output (*SNPO*) compared to HCs. Bianchi et al. investigated the environmental impact of hydrofluoroolefines (HFOs) for micro-ORCs. Despite having very low GWP compared to HFCs, HFOs are found to be less efficient than HFCs.

It is observed that most of the ORCs investigated and reported in the literature are simple Rankine cycles, i.e., with no attempt to study the possibility of internal heat recovery. Moreover, most ORC studies found are with sink temperatures in the range of 10 to 25 °C. The current study aims to evaluate the performance of ORC for a warm region with a condenser temperature of 40 °C for source temperatures varying from 150 to 300 °C. ORC with internal heat recuperation is investigated for 12 different organic fluids (mostly HCs). The study aims to provide the best candidates (organic fluids) for a given source temperature and evaluate the need for an internal heat recovery unit in the system for each case. Additionally, the size of the heat exchangers is also estimated by calculating the conductance (UA) of the heat exchangers in the cycle.

## 2. Selection of Organic Fluids and Objectives

Utilization of organic fluids in ORC has constraints such as limited temperature ranges, toxicity, and environmental impact. The selection of organic fluids to operate ORC may also be influenced by its market availability and cost. One must take care of safety measures to mitigate related issues. Optimal fluid selection and control strategies may help tackle these challenges to some extent. For the sustainable operation of ORCs, it is crucial to have a comprehensive grasp of the limitations and implement suitable mitigation strategies, despite the assistance provided by organic fluids in harnessing low-grade heat.

A significantly wide variety of organic fluids is available, which can be used as working fluid in the organic Rankine cycle (ORC). A wide range of organic fluids to choose from also makes the selection complicated for a given source temperature. Organic fluids selected for the current investigation are mostly hydrocarbons as they are generally considered good candidates for medium-temperature waste heat recovery and geothermal applications [27–29]. Table 1 lists the thermodynamic properties of 12 organic fluids considered to study their performance in the ORC. The selection of the organic fluid to be used in the Rankine cycle for a given source temperature significantly affects its overall performance, which includes the specific net power output of the cycle, energy, and exergy efficiencies [5,30,31]. This is due to varying thermophysical properties of organic fluids, such as critical temperature, specific heat, latent heat of vaporization, and normal boiling temperature. The current study targets the selection of organic fluids suitable for a heat source temperature in the range of 150 to 300 °C; therefore, selected organic fluids have their critical temperatures in the same range.

**Table 1.** List of organic fluids used to operate ORC in the current study.

| | Name | Chemical Formula/Alternative Name | Critical Pressure (MPa) | Critical Temperature (°C) | Normal Boiling Temperature (°C) |
|---|---|---|---|---|---|
| 1 | Pentane | C5H12 | 3.4 | 197 | 36.1 |
| 2 | Cyclopentane | C5H10 | 4.5 | 239 | 49.2 |
| 3 | Cyclohexane | C6H12 | 4.1 | 281 | 80.8 |
| 4 | Acetone | C3H6O | 4.9 | 235 | 56.1 |
| 5 | n-Butane | R600 | 3.8 | 152 | −0.5 |
| 6 | CIS-Butene | - | 4.2 | 162 | 3.7 |
| 7 | RE347mcc | HFE-7000 | 2.5 | 165 | 34.2 |
| 8 | Neopentane | C5H12 | 3.2 | 161 | 9.5 |
| 9 | Benzene | C6H6 | 4.9 | 289 | 80.1 |
| 10 | Isopropyl Alcohol | C3H8O | 5.4 | 236 | 82.5 |
| 11 | Hexafluorobenzene | C6F6 | 3.3 | 244 | 80.1 |
| 12 | R245fa | C3H3F5 | 3.7 | 154 | 15.3 |

## 3. Description of ORC Configuration

Our current investigation focuses on studying the organic Rankine cycle with internal heat recuperation. Figure 1a provides a visual representation of the cycle's layout, which includes essential components such as a pump, an evaporator, an expander, a condenser, and an internal heat recovery unit (IHRU). The organic fluid is pumped into the evaporator, where it absorbs heat from a stream of hot air entering the system. Subsequently, the high-temperature and high-pressure fluid undergoes expansion in the expander. During the condensation process, the IHRU recovers heat from the stream exiting the expander, ensuring that it is not wasted but rather released into the seawater.

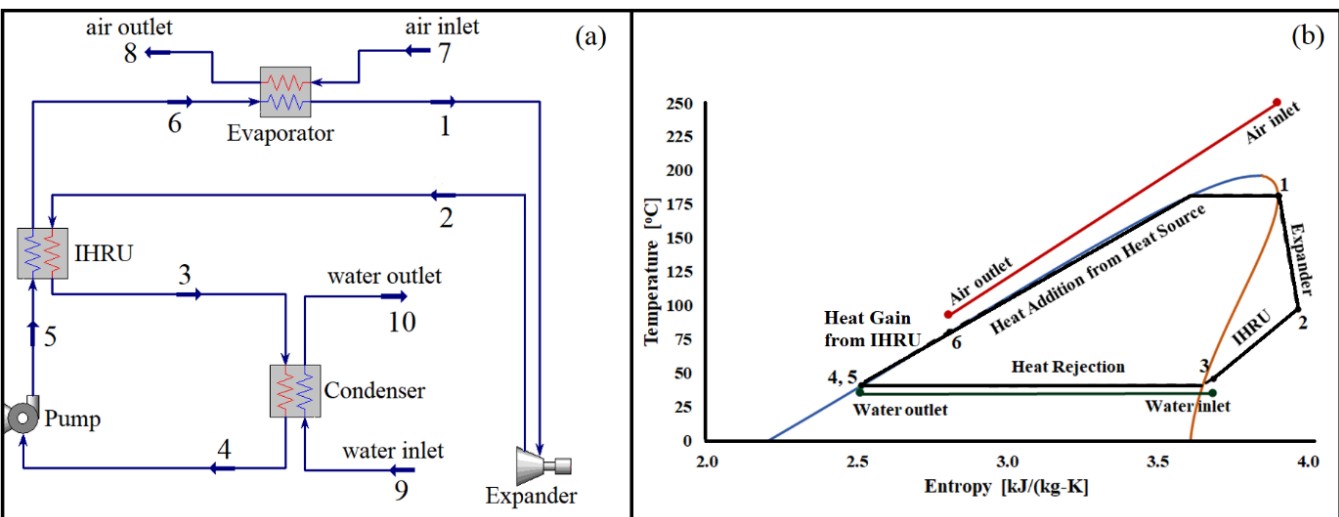

**Figure 1.** (**a**) Organic Rankine cycle with an internal heat recovery unit and (**b**) T-s diagram of the cycle using pentane as a working fluid operating with a source temperature of 250 °C. Numbers (1 to 10) represent state points. In heat exchangers (Evaporator, Condenser, and IHRU) red and blue lines represent hot-side and cold-side streams, respectively.

To further illustrate the cycle's operation, Figure 1b presents a temperature-entropy diagram at a specific source temperature. The expander generates power output as the working fluid expands between state points 1 and 2. Notably, the temperature of the fluid at state 2 is significantly higher than the ambient temperature. Without the implementation of the IHRU, the heat from the stream leaving the expander would be lost. However, by utilizing the IHRU, the heat from the stream between state points 2 and 3 is effectively recovered, leading to an increase in the working fluid's temperature between state points 5 and 6. Finally, the stream between state points 6 and 1 receives heat from the source (hot air) to reach the desired operating temperature of the cycle.

## 4. Methodology and Operating Parameters

The cycle depicted in Figure 1a is simulated using Aspen HYSYS V11. Thermophysical properties of the working fluids are determined using the Peng-Robinson model. An overview of the investigated working fluids can be found in Table 1. The cycle is assumed to operate under steady-state conditions, with no losses of pressure or heat in the connecting pipelines that link the cycle components. Furthermore, pressure drops in the evaporator, condenser, and heat recuperator (for both cold and hot streams) are neglected.

The objective of this study is to analyze the exergetic performance of various organic fluids in an organic Rankine cycle (ORC) designed for warm regions, focusing on a specific source temperature. Therefore, the dead-state temperature and pressure, representing ambient conditions, are assumed to be 35 °C and 101 kPa, respectively. Only subcritical ORC configurations are considered, and the maximum allowable cycle pressure is set not to exceed 90% of the critical pressure of the working fluid. To maintain optimal exergy efficiency, it is known that introducing superheat to the expander inlet decreases performance [32,33]. Hence, the expander inlet temperature is adjusted to the saturation temperature corresponding to the evaporator pressure.

To prevent negative pressure within the cycle, the minimum allowable cycle pressure is established at 101 kPa. Negative pressure could lead to the intake of atmospheric air in the event of leaks. In the evaporator, the working fluid is heated by hot air with inlet temperatures ranging from 150 to 300 °C. A minimum allowable pinch point temperature difference (PPTD) of 10 °C is maintained during this process. During condensation, water at an inlet temperature of 35 °C acts as the heat sink for the working fluid, with a minimum

allowable PPTD of 5 °C. Details regarding the design and operating parameters of the cycle can be found in Table 2.

**Table 2.** Designed parameters and values used in modeling ORC [33].

| Parameter | Value |
|---|---|
| Heating medium (source) | Air with inlet temperature 150–300 °C and 101 kPa at 1 kg/s |
| Cooling medium (sink) | Water at 35 °C and 101 kPa |
| Dead state condition | 35 °C and 101 kPa |
| Expander inlet pressure | Up to 90% of critical pressure of the working fluid |
| Expander inlet temperature | Saturation temperature corresponding to expander pressure |
| Pump inlet temperature | 40 °C |
| Pump inlet pressure | 101 kPa or saturation pressure at 40 °C if it is above 101 kPa |
| Pump adiabatic efficiency | 0.9 |
| Expander adiabatic efficiency | 0.8 |
| Minimum PPTD (evaporator) | 10 °C |
| Minimum PPTD (condenser) | 5 °C |
| Minimum PPTD (IHRU) | 5 °C |

## 5. Performance Analysis of ORC

This section highlights the essential equations used in the analysis to evaluate the performance of the cycle. There is no doubt that the energetic performance of the cycle plays an important role in quantifying its effectiveness; however, exergy analysis provides a finer understanding of the energy losses arising in the system due to irreversibilities. The higher exergy efficiency of the cycle reflects greater utilization of available resources and therefore makes the system more sustainable. Therefore, the current work focuses on the exergetic performance of the cycle.

The net exergy flow rate ($E_i$), at any given state point ($i$) of the cycle shown in Figure 1a, is calculated as:

$$E_i = \dot{m}_i(h_i - T_a s_i) \tag{1}$$

where $h_i$ is the enthalpy [kJ/kg], $s_i$ is the entropy [kJ/kg K], and $m_i$ is the mass flow rate [kg/s] of the working fluid. $T_a$ represents the dead-state temperature in Kelvin, which is equal to the ambient temperature.

Exergy losses due to irreversibilities in each component of the cycle are computed as:

$$I_{expander} = (E_1 - E_2) - W_E \text{ [kW]} \tag{2}$$

$$I_{pump} = W_P - (E_5 - E_4) \text{ [kW]} \tag{3}$$

$$I_{evaporator} = (E_7 - E_8) - (E_1 - E_6) \text{ [kW]} \tag{4}$$

$$I_{IHRU} = (E_2 - E_3) - (E_6 - E_5) \text{ [kW]} \tag{5}$$

$$I_{condenser} = (E_3 - E_4) - (E_{10} - E_9) \text{ [kW]} \tag{6}$$

In Equations (2) and (3), $W_E$ and $W_P$ represents the power output and power input of the expander and pump, respectively. It is to be noted that the net exergy loss ($E_{net\_loss}$) can be obtained by adding up all irreversibilities obtained from Equations (2)–(6). Finally, the exergy efficiency of the cycle is computed as:

$$\eta_X = (1 - E_{net\_loss}/E_{in}) \times 100 \text{ [%]} \tag{7}$$

where $E_{in}$ is the net exergy provided to the cycle, which is equal to

$$E_{in} = E_7 - E_8 \text{ [kW]} \tag{8}$$

## 6. Results and Discussion

This section describes the essential results of the study. Various operating parameters affecting the performance of the cycle will be investigated. The exergetic performance of the organic fluids in the cycle at various source temperatures shall be compared and discussed.

### 6.1. Optimal Evaporator Pressure

Evaporator and condenser pressures play an important role in the overall performance of the cycle. In the current investigation, the minimum condensing pressure is fixed, which is assumed to be 101 kPa, or the saturation pressure corresponding to the minimum temperature of the cycle (i.e., 40 °C) in the case it is more than 101 kPa. Therefore, the exergy efficiency is evaluated for the cycle at various evaporator pressures. Please note that no pressure losses are considered in the current study; therefore, the evaporator pressure is the same as the expander inlet pressure, and the condenser pressure is equal to the pump inlet pressure. Figure 2 presents the plots of exergy versus expander inlet pressure at two source temperatures (i.e., 150 and 250 °C). The organic fluids for each case were selected randomly to investigate the general pattern. It is clear from these plots that, irrespective of source temperature and organic fluid, the exergy efficiency increases nonlinearly with increasing expander inlet pressure to a maximum value, and then it either declines slightly or shows not much appreciation. Therefore, for each case, the expander inlet pressure is set, which maximizes the exergetic performance of the cycle.

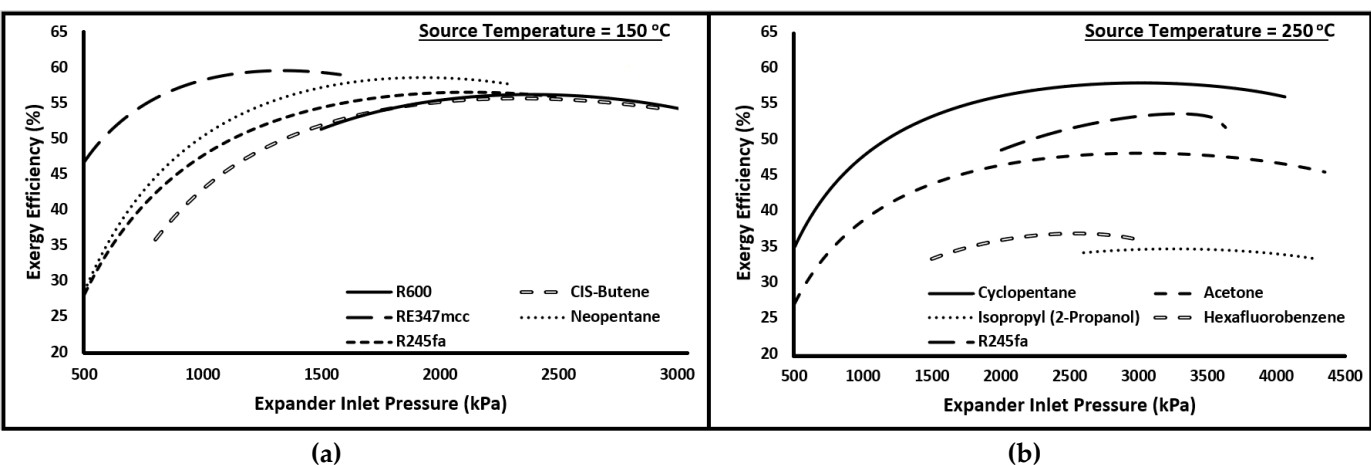

**(a)**                                                                                                 **(b)**

**Figure 2.** Exergy efficiency versus Expander inlet pressure for (a) R600, CIS-Butene, RE347mcc, Neopentane, and R245fa at source temperature of 150 °C and (**b**) Cyclopentane, Acetone, Isopropyl (2-Propanol), Hexafluorobenzene, and R245fa at source temperature of 250 °C.

### 6.2. Effect of the Source Temperature on the Exergy Performance of the Cycle

The source temperature is one of the important parameters in the selection of organic fluid for ORC. This section discusses the exergy performance of the cycle for the organic fluids selected in this study at four different source temperatures from 150 to 300 °C. For each source temperature, the exergy efficiency of the cycle is obtained by operating at optimal evaporator pressure.

Figure 3 presents the plots of exergy efficiencies at source temperatures 300, 250, 200, and 150 °C. The expander inlet temperatures (black dots) are also plotted along with the critical temperature (red diamonds) of the fluid. This plot shows a very interesting feature between how the exergy efficiency varies with increasing source temperature in relation to the expander inlet temperature and critical temperature. For example, considering

pentane at a source temperature of 150 °C, the exergy efficiency is 61.7% and the difference between its critical temperature and expander inlet temperature is nearly 57 °C (distance between red diamond symbol and black dot). Increasing the source temperature from 150 to 200 °C improves the exergy efficiency but it reduces the distance between the red diamond symbol and black dot. The improvement in exergy efficiency continues till the difference between the critical temperature of the fluid and expander inlet temperature continue to drop; thereafter, further increases in source temperature negatively impact the exergetic performance of the cycle. This behavior is similar for all organic fluids. Like, for n-Butane, CIS-Butene, RE347mcc, Neopentane, and R245fa, increasing the source temperature beyond 200 °C reduces the exergy efficiency.

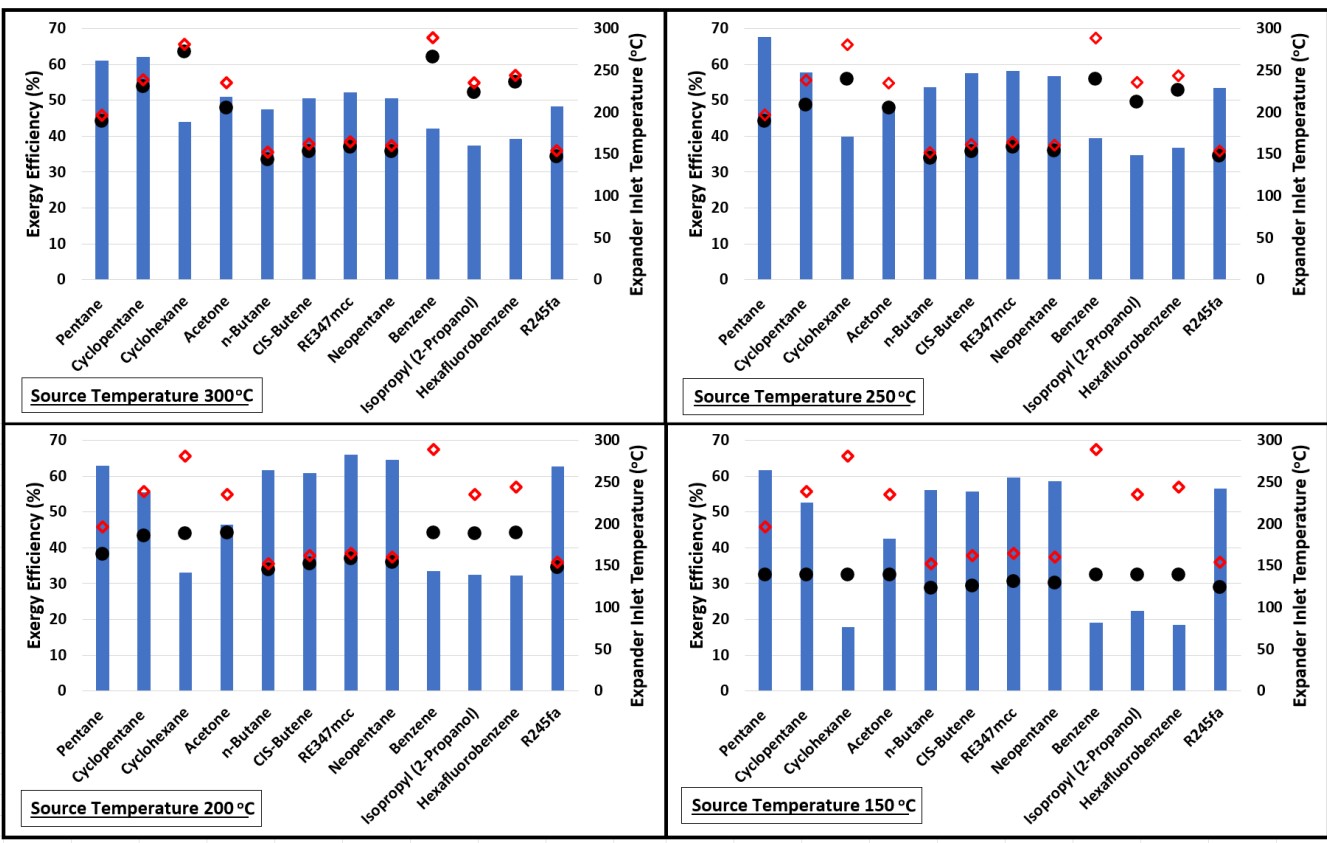

**Figure 3.** Exergy performance of the cycle using various organic fluids at source temperatures of 150, 200, 250, and 300 °C. Red diamonds and black dots represent the critical temperature of the fluid and expander inlet temperature, respectively.

The change in the exergy efficiency (in percentage points) between each source temperature is calculated and plotted in Figure 4. A big improvement in the exergy efficiency (nearly 15% points) is noticed when the source temperature is increased from 150 to 200 °C for cyclohexane, benzene, isopropyl alcohol, and hexafluorobenzene. However, these four are the least-performing organic fluids in comparison to others at all source temperatures, as seen in Figure 3. One of the reasons for their poor performance is their normal boiling temperatures, which are significantly higher compared to the minimum temperature of the organic fluid in the cycle (i.e., 40 °C). Therefore, the working fluid cannot expand to the saturation pressure corresponding to 40 °C in the expander as the minimum pressure the fluid can expand to is 101 kPa to avoid a vacuum in the system. As a result, a substantially large portion of the exergy loss incurs in the condenser for cyclohexane, benzene, isopropyl alcohol, and hexafluorobenzene, as evident from Figure 5. This figure shows the exergy losses incur in each component of the cycle.

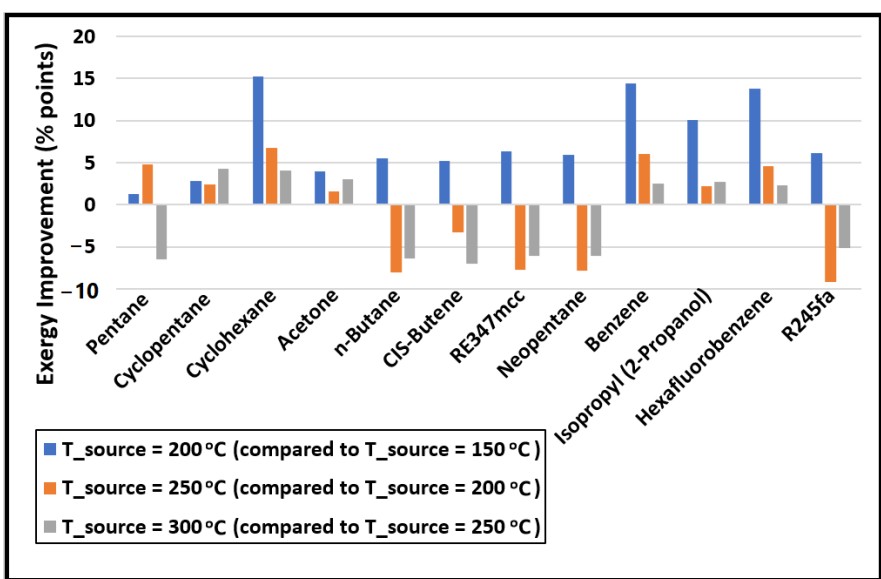

**Figure 4.** The change in the exergy efficiency (in percentage points) of the cycle with increasing source temperature.

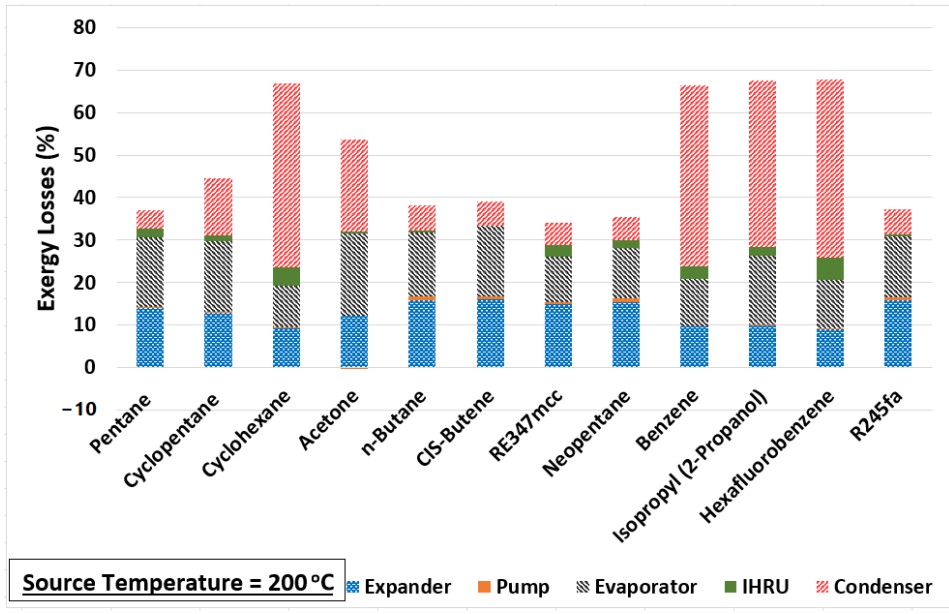

**Figure 5.** Component-wise exergy loss in the cycle at source temperature of 200 °C.

Increasing the source temperature above 200 °C for n-Butane, CIS-Butene, RE347mcc, neopentane, and R245fa reduces the exergy efficiency. This is mainly due to their low critical temperatures; therefore, any further increase in the source temperature hurts the exergy efficiency of the cycle. For these fluids, on average, nearly 8% points in exergy efficiency are dropped for each 50 °C increase in the source temperature. Table 3 lists the values of the exergy efficiency of the cycle for each organic fluid. This table provides quick insight into which organic fluid (among the selected ones in this study) is the best for a given source temperature.

**Table 3.** Numerical values of the exergy efficiency (in percentage) of the cycle obtained for various organic fluids at different source temperatures.

| Organic Fluid | Source Temperature | | | |
|---|---|---|---|---|
| | 150 °C | 200 °C | 250 °C | 300 °C |
| Pentane | 61.7 | 62.9 | 67.7 | 61.2 |
| Cyclopentane | 52.6 | 55.5 | 57.9 | 62.2 |
| Cyclohexane | 17.9 | 33.2 | 39.9 | 44.0 |
| Acetone | 42.5 | 46.5 | 48.1 | 51.1 |
| n-Butane | 56.2 | 61.7 | 53.8 | 47.4 |
| CIS-Butene | 55.7 | 60.9 | 57.6 | 50.6 |
| RE347mcc | 59.6 | 65.9 | 58.2 | 52.1 |
| Neopentane | 58.6 | 64.5 | 56.7 | 50.7 |
| Benzene | 19.1 | 33.5 | 39.5 | 42.1 |
| Isopropyl (2-Propanol) | 22.4 | 32.5 | 34.7 | 37.4 |
| Hexafluorobenzene | 18.5 | 32.3 | 36.9 | 39.2 |
| R245fa | 56.6 | 62.7 | 53.5 | 48.4 |

*6.3. Effect of the Organic Fluid on Specific Net Power Output of the Cycle*

To assess the capability of power generation of each organic fluid at various source temperatures, the specific net power output (*SNPO*) of the cycle is calculated using:

$$SNPO = (W_E - W_P)/\dot{m}_{wf} \ [\text{kJ/kg}] \tag{9}$$

where $\dot{m}_{wf}$ is the mass flow rate of the working fluid in the cycle. Figure 6 shows the values of *SNPO* plotted for various organic fluids used. Based on Table 3, cyclohexane, benzene isopropyl alcohol, and hexafluorobenzene are not considered due to their poor exergy performance. Figure 6 shows the values of *SNPO* plotted for the organic fluids with exergy efficiency above 50%. It is evident from this figure that RE347mcc and R245fa provide the least specific net power output (nearly 30 kJ/kg on average); therefore, for a given power net output, a higher mass flow rate is needed for RE347mcc and R245fa in comparison to other organic fluids. Increasing the mass flow rate requirements may lead to the increase in size of the turbomachinery (expander and pump) and the heat exchangers. Acetone presents the highest *SNPO*, but its exergy performance is significantly low in comparison to the rest. On the other hand, pentane being highly efficient also provides reasonably good *SNPO* at all source temperatures (from 75 to 95 kJ/kg). n-butane, CIS-butene, and neopentane behave nearly similarly giving out *SNPO* of nearly 60 kJ/kg.

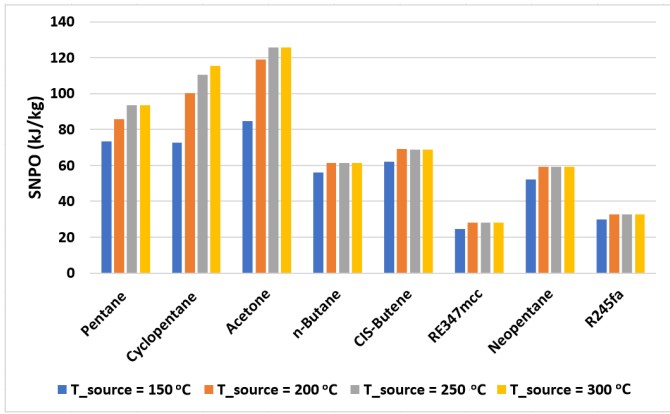

**Figure 6.** The specific net power output of the cycle with selected organic fluids operating at various source temperatures.

*6.4. Performance of the Internal Heat Recovery Unit in the Cycle*

We have seen in the previous sections that the exergy performance of the cycle is strongly dependent on the organic fluid used and source temperature. This section is

dedicated to highlighting the importance and role of the internal heat recovery unit (IHRU) in the cycle. The job of IHRU in the cycle is to recover the heat from the stream leaving the expander to save and reduce the amount of heat needed at the evaporator. However, not all organic fluids behave the same; therefore, to assess the effectiveness of IHRU in the cycle, the heat energy recovered by IHRU with respect to the net heat energy needed to run the cycle is calculated. It is obtained using:

$$IHRU\ Contribution = \left( \frac{Q_{IHRU}}{Q_{IHRU} + Q_{Evap}} \right) \times 100\ [\%] \tag{10}$$

where $Q_{IHRU}$ represents the amount of heat gain by the cold stream of the IHRU (between state points 5 and 6, refer to Figure 1) and $Q_{Evap}$ is the heat gain by working fluid in the evaporator (between state points 6 and 1). Due to the poor exergy performance of the cycle with cyclohexane, benzene isopropyl alcohol, and hexafluorobenzene, they are not included in this analysis. It is worth recalling that the pump inlet temperature is fixed to 40 °C; therefore, if the temperature of the stream leaving expander is significantly higher than 40 °C and the heat is not recovered from it, then it is rejected in the condenser, which ultimately affects the exergy performance of the cycle.

Figure 7 presents the IHRU contribution in the cycle calculated using Equation (10); symbols plotted in this figure represent the exergy efficiencies of the cycle at various source temperatures. It is observed that the IHRU role is significantly higher in recuperating heat from the cycle when operating with pentane, cyclopentane, RE347mcc, and neopentane. Pentane was found to be highly efficient (with exergy efficiency above 60%) at all source temperatures. It showed the highest exergy efficiency of 67.7% at a source temperature of 250 °C for which the IHRU contribution is nearly 17%. Acetone is the least efficient and needs a heat recuperation cycle with a contribution of the IHRU of nearly 7.5%; this means its performance would further decline if a simple Rankine cycle were used instead of a heat recuperation cycle. The cycle operating with CIS-butene doesn't need the IHRU as it shows nearly no contribution in recovering the heat. This means that the CIS-butene would perform better in comparison to other organic fluids if the simple (standard) Rankine cycle is used n-butane, R245fa offers exergy efficiency of nearly 62% at a source temperature of 200 °C with an IHRU contribution of nearly 5%; this implies that in comparison to pentane, cyclopentane, RE347mcc, and neopentane, these two organic fluids would perform better if a simple Rankine cycle were used.

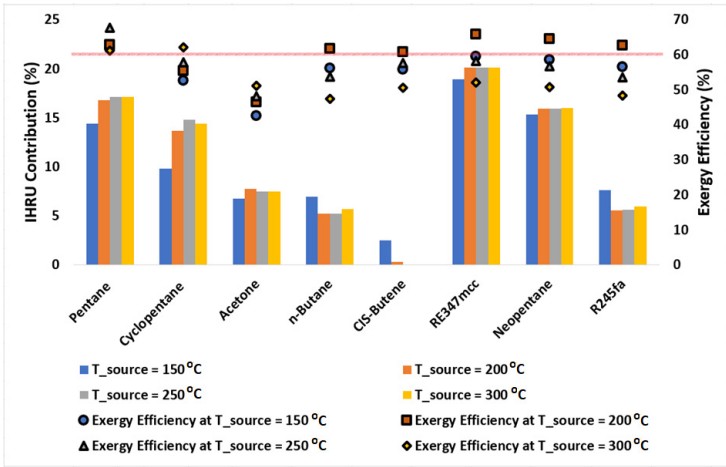

**Figure 7.** Column chart showing the contribution of the IHRU in the cycle operating at source temperatures of 150, 200, 250, and 300 °C. Symbols represent the exergy efficiency of the cycle. A solid horizontal line is plotted for a reference purpose to separate data points with exergy efficiency above 60%.

*6.5. Heat Exchangers*

This section assesses the sizes of the heat exchangers in the system. The size of the evaporator, condenser, and IHRU is evaluated by calculating the heat exchanger conductance (i.e., $UA$ values). The $UA$ value of the heat exchanger plays an important role in the selection of its type for a given application. For each heat exchanger in the system, it is calculated as:

$$UA = Q_{duty}/LMTD \; [\text{kW/K}] \tag{11}$$

where $Q_{duty}$ is the heat exchanger duty and $LMTD$ is the logarithmic mean temperature difference.

The net heat exchanger size (net conductance, $UA_{net}$) for the system can be expressed as:

$$UA_{net} = UA_{Evap} + UA_{Cond} + UA_{IHRU} \; [\text{kW/K}] \tag{12}$$

where $UA_{Evap}$, $UA_{Cond}$, and $UA_{IHRU}$ represent the conductance of the evaporator, condenser, and IHRU, respectively. It is worth recalling that the mass flow rate of the organic fluid in the cycle is not equal for each case. Only the mass flow rate of the heating medium (i.e., hot air) is taken constant, which is 1 kg/s; therefore, to compare the size of the system, the net conductance ($UA_{net}$) of the system for each case is normalized by the net power output of the cycle.

Figure 8 represents the conductance normalized by the net power output of the cycle for the evaporator, condenser, and IHRU. Irrespective of the source temperature and organic fluid, we observe that the size of the IHRU is significantly smaller than the evaporator and the condenser. This signifies that the size of the heat exchanger used in heat recovery is relatively smaller than other heat exchangers in the system. As observed in the previous sections, the IHRU plays a significant role in improving the overall exergy performance; therefore, the current study supports the use of ORC with internal heat recuperation when compared with standard ORC.

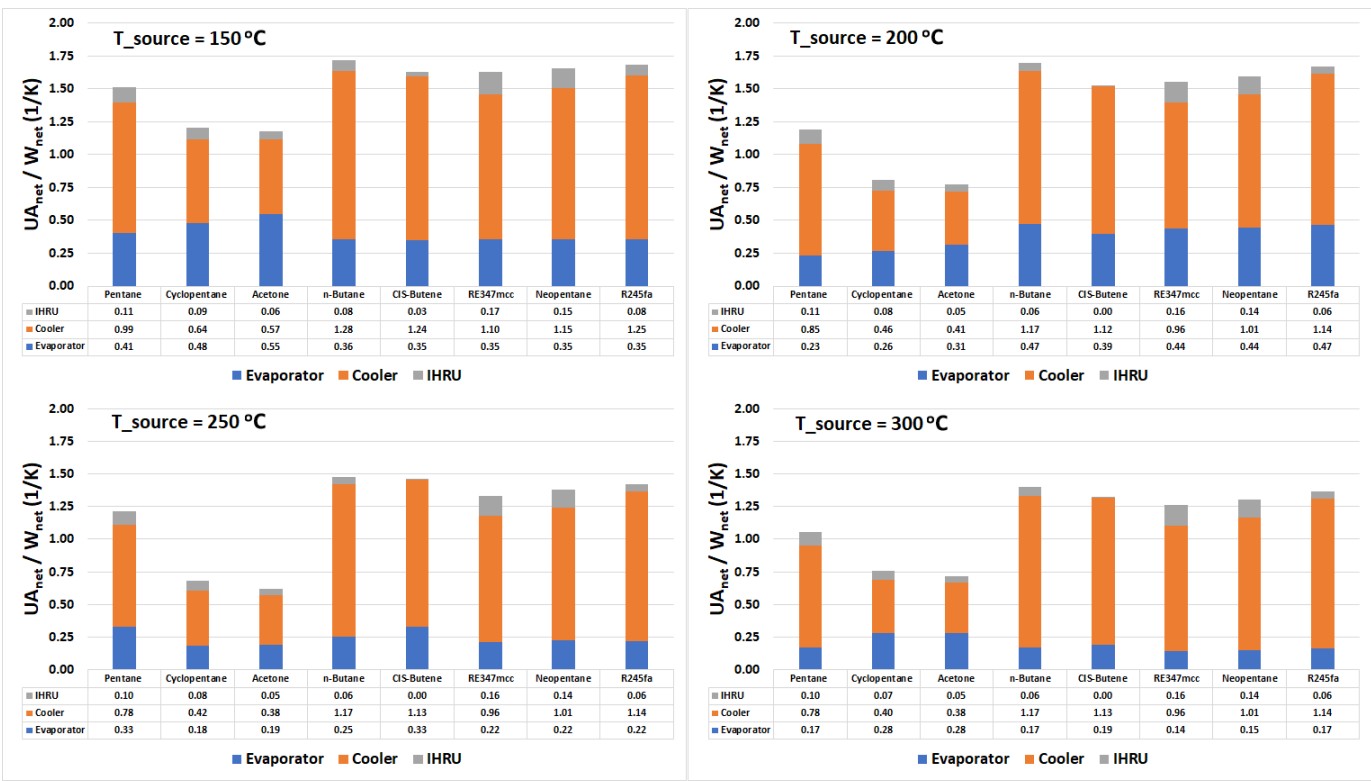

**Figure 8.** Stacked column chart displaying the conductance ($UA$ value) of the evaporator, cooler, and IHRU in the cycle.

Acetone is the worst performer in the exergy analysis among selected organic fluids (see Figure 7), so it is disregarded in the discussion. At a source temperature of 300 °C, cyclopentane is found to be the best performer in terms of the exergy efficiency, and the net heat transfer area needed per unit net power output of the cycle. Considering the source temperature of 250 °C, pentane is the most exergy efficient and outperforms in comparison to other organic fluids; however, it needs a larger heat transfer area when compared with cyclopentane. At 200 °C source temperature, RE347mcc and neopentane performed relatively better than other organic fluids. Figure 8 suggests that they both require nearly the same heat transfer area. However, on the other hand, pentane being only 3% points less exergy efficient than RE347mcc needed significantly less heat transfer area. At a source temperature of 150 °C, the three best performers in terms of exergy efficiency and required net heat exchanger size are pentane, RE347mcc, and neopentane. However, pentane still has an upper hand as it is more exergy efficient with slightly less heat transfer area needed.

## 7. Conclusions

The organic Rankine cycle with an internal heat recovery unit (IHRU) was investigated for various organic fluids. The cycle was assessed for its exergy performance at source temperatures varying from 150 to 300 °C. The study considered 12 different organic fluids, chosen for their thermophysical properties—particularly their critical temperatures—which match the source temperature range under investigation. These are pentane, cyclopentane, cyclohexane, acetone, n-butane, CIS-butane, RE347mcc, neopentane, benzene, isopropyl alcohol (2-propanol), hexafluorobenzene, and R245fa. The exergetic performance and specific net power output of the ORC are evaluated for each organic fluid operating at different source temperatures. To assess the potential improvement in the performance of the cycle with IHRU, the degree of enhancement in the cycle's exergy performance was assessed. Below is a summary of the key results of the current work.

- The critical temperature and pressure of an organic fluid play an important role in its selection for ORC at a given source temperature.
- Cyclohexane, benzene, isopropyl alcohol, and hexafluorobenzene performed very poorly in terms of exergy efficiency when compared to other organic fluids. In comparison to cycle minimum temperature, i.e., 40 °C, these four organic fluids have a boiling point considerably higher. This restricted the pressure to which the working fluid expanded in the expander, which ultimately increased the exergy losses in the condenser.
- At a source temperature of 300 °C, pentane, and cyclopentane outperformed the other organic fluids, whereas cyclopentane (62.2%) had slightly better exergy efficiency than pentane (61.2%). Moreover, Cyclopentane had substantially better SNPO than pentane (nearly 22 kW/kg higher). Also, the net heat exchange area (UA) needed per unit net power output of the plant was considerably smaller for cyclopentane than pentane, this may lead to a smaller plant size.
- At a source temperature of 250 °C, pentane was found to be the best candidate in terms of exergy efficiency followed by RE347mcc and cyclopentane. The heat transfer area needed for RE347mcc was comparable to pentane; however, specific net power output was too low compared to pentane and cyclopentane. Cyclopentane was slightly less efficient than pentane but had significantly higher SNPO and lower UA value. Therefore, cyclopentane is still an attractive option for a source temperature of 250 °C.
- At a source temperature of 200 °C, RE347mcc achieved the highest exergy efficiency followed by neopentane and pentane. However, neopentane and RE347mcc needed higher heat exchanger sizes with substantially lower SNPO values in comparison to pentane. Pentane being 3 percentage points less exergy efficient than RE347mcc is still a good option due to SNPO and UA values.

- At a source temperature of 150 °C, the three best candidates in terms of exergy efficiency were pentane, RE347mcc, and neopentane. However, pentane is still superior due to considerably higher SNPO, and lower UA value compared to RE347mcc and neopentane.

**Author Contributions:** Conceptualization, M.E.S. and E.A.; Methodology, M.E.S. and E.A.; Software, M.E.S.; Investigation, M.E.S. and E.A.; Resources, M.E.S.; Data curation, U.S. and A.A.T.; Writing–original draft, M.E.S.; Writing–review & editing, E.A., U.S. and A.A.T.; Visualization, U.S.; Project administration, A.A.T.; Funding acquisition, M.E.S. All authors have read and agreed to the published version of the manuscript.

**Funding:** This research work was funded by Institutional Fund Projects under grant no. (IFPIP: 771-135-1443). The authors gratefully acknowledge technical and financial support provided by the Ministry of Education and King Abdulaziz University, DSR, Jeddah, Saudi Arabia.

**Conflicts of Interest:** The authors declare no conflict of interest.

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
