# Peer review of "Selection of Organic Fluid Based on Exergetic Performance of Subcritical Organic Rankine Cycle (ORC) for Warm Regions"

_energies, doi:10.3390/en16135149_

Round 1

Reviewer 1 Report

(Abstract)
- Please provide more specific findings or quantitative results.
(Introduction)
- The innovation of this work should be further enhanced. The literature review lacks critical comments. More current papers published on Engergies may be cited, such as https://doi.org/10.3390/en15176230.

(Selection of Organic Fluids and Objectives)
- The low-GWP HFOs fluids may be considered in this work.
(Methodology and Operating Parameters)
- A flowchart may be useful for understanding.
(Results and discussion)
- How did you optimize the evaporator pressure?

The manuscript was well written.

Author Response

Thanks for your comments and we appreciate the time you spend in carefully reading the manuscript.

AS per your suggestions, we have modified the abstract and included specific findings and values of exergy efficiencies for each case. 

Moreover, we have also included the suggested citation in the revised version.

Thanks for suggesting  utilization of low-GWP HFOs fluids. This work in still in progress and in future work we will include these fluids along with there zeotropic mixtures.

For the optimization of evaporator pressure please refer to section 6.1 and Figure 2. For each case, we find the optimal evaporator pressure which maximizes the exergy efficiency.

Hope we have answered your question.

Reviewer 2 Report

The author selected a heat source temperature of 150-300 and analyzed the exergy efficiency and heat transfer area of the indirect system using 12 working fluids. The main questions and suggestions:

1The author selected the heat source temperature of 300 oC for calculation, and among the 12 selected working fluids, some of them had a critical temperature below 200 oC. Please supplement the efficiency data of the main equipment to facilitate the determination of the distribution of system exergy losses.

2The author mentioned that: with no attempt to study the possibility of internal heat recovery. There are many studies on the internal heat recovery system. Please clarify which aspects of the internal heat recovery system have not been studied, such as working fluids, research methods, or temperature of cold and heat sources.

3Please write clearly what the black and red symbols representfor example Figure 3.

4Is the SNPO method proposed by the author or by other scholars? What are the advantages of SNPO?

5The article proposes 12 working fluids, but why only some of them are calculated in Figure 2

6The suggestion is to change the form of Table 3 and Fig 3.-8. to a curve chart for easy comparison and analysis of datas.

English Language is acceptable.

Reviewer 3 Report

- Rearrange keywords alphabetically.

- In the introduction section, the objectives of the research should be stated one by one.

- Why you explained figure 8 before figure 7?

- Please do not miss "°" in the legend of the figures.

Moderate editing of English language required

Reviewer 4 Report

The manuscript "Investigation of the Organic Rankine Cycle with an Internal Heat Recovery Unit Using Various Organic Fluids" is a comprehensive and highly informative research work, providing insights into the exergy performance and specific net power output of an Organic Rankine Cycle (ORC) under various conditions. The authors have adopted a well-structured approach to investigate the performance of twelve different organic fluids, considering the impact of the source temperature and the inclusion of an internal heat recovery unit (IHRU).

The selection of organic fluids and the range of source temperatures are well justified and cover a wide spectrum of conditions. The authors have provided a thorough analysis of the results and have identified the top performing fluids under various scenarios, which could guide researchers and practitioners in selecting the appropriate fluid for their specific applications.

The authors have addressed the critical role of the IHRU in the cycle, improving the exergy efficiency and highlighting its significantly smaller size compared to the Evaporator and the Condenser. This advocates for the use of an ORC with IHRU rather than a standard ORC, especially when operating with fluids like Pentane, RE347mcc, Cyclopentane, and Neopentane.

However, there are a few aspects that require further attention:

  1. The manuscript doesn't touch upon the environmental, safety, and economic aspects of the organic fluids, which are important factors for consideration in real-world applications. Including these aspects could provide a more comprehensive analysis and make the work more useful for practitioners.
  2. The paper is mostly focused on the theoretical aspect. Including some experimental data or validation could enhance the credibility and impact of the work.
  3. It would be beneficial to expand the discussion on the limitations and potential drawbacks of using these organic fluids in ORCs, as well as potential mitigations or alternative solutions.
  4. The implications of these results for different types of applications (for example, power generation, heating, cooling, etc.) could be discussed in more detail.

Overall, this is a solid research paper with valuable insights for the field of ORC. With the addition of these suggestions, the manuscript could provide a more well-rounded analysis, making it more beneficial to a wider audience.

The English language used in the paper is generally clear and understandable. However, there are a few areas where the phrasing could be improved for clarity and readability. For instance:

"Twelve different organic fluids were considered in the study due to their thermophysical properties, especially their critical temperatures, to match the source temperature range considered in the current study." - This sentence could be rewritten to avoid repetition and improve clarity. An alternative could be: "The study considered twelve different organic fluids, chosen for their thermophysical properties - particularly their critical temperatures - which match the source temperature range under investigation."

"The extent of improvement in the exergy performance of the cycle with IHRU was evaluated to investigate its potential and need in the cycle." - The phrase "its potential and need in the cycle" is somewhat unclear. Consider specifying what "its" is referring to (the IHRU or the improvement in exergy performance) and what is meant by "potential and need".

There are places where abbreviations are used before being defined, for example "SPNO". Always define an abbreviation the first time you use it.

In the conclusions section, the term "net specific power output" is used in some places and "specific net power output" in others. Consistency is important for clarity.

Proofreading for such inconsistencies and areas of potential confusion would be beneficial before final submission. Otherwise, the English language usage in the paper is of a good standard.

Round 2

Reviewer 2 Report

Please make corresponding modifications in the article. 

Quality of English Language is ok.